# Direct observation of a few-photon phase shift induced by a single quantum emitter in a waveguide

Mathias J. R. Staunstrup[1], Alexey Tiranov [1], Ying Wang[1], Sven Scholz[2], Andreas D. Wieck [2], Arne Ludwig [2], Leonardo Midolo [1], Nir Rotenberg [1], Peter Lodahl [1] ✉ & Hanna Le Jeannic [1] ✉

Realizing a sensitive photon-number-dependent phase shift on a light beam is required both in classical and quantum photonics. It may lead to new applications for classical and quantum photonics machine learning or pave the way for realizing photon-photon gate operations. Nonlinear phase-shifts require efficient light-matter interaction, and recently quantum dots coupled to nanophotonic devices have enabled near-deterministic single-photon coupling. We experimentally realize an optical phase shift of $0.19\pi \pm 0.03$ radians ($\approx 34$ degrees) using a weak coherent state interacting with a single quantum dot in a planar nanophotonic waveguide. The phase shift is probed by interferometric measurements of the light scattered from the quantum dot in the waveguide. The process is nonlinear in power, the saturation at the single-photon level and compatible with scalable photonic integrated circuitry. The work may open new prospects for realizing high-efficiency optical switching or be applied for proof-of-concept quantum machine learning or quantum simulation demonstrations.

Optical nonlinearities are at the core of many modern applications in photonics. If sensitive at the level of single light quanta, they may be applied to realize fundamental quantum gate operations for photonic quantum computing or advanced quantum network implementations[1,2]. The nanophotonics platform could potentially be scaled up to realize large-scale nonlinear quantum photonic circuits, as required, e.g., in quantum neural networks[3]. Strong optical nonlinearities can be achieved using single emitters such as molecules or quantum dots (QDs) embedded in photonic waveguides or cavities[4,5] due to the tight confinement of light to reach light-matter coupling efficiencies near unity[6]. In the waveguide geometry, a narrow-band single-photon wavepacket is deterministically reflected upon resonant interaction with a highly coherent two-level quantum emitter, while two-photon wavepackets are partly transmitted due to the saturation of the emitter[7,8], allowing for realizing deterministic quantum operations such as photon sorters[9,10]. In contrast to optimal $\pi(\pi/2)$ phase

shift operations, even moderate nonlinear interactions have been proposed as a way to boost measurement-based quantum computing[11] and for the implementation of quantum neural networks[3,12].

Emitter-induced phase shifts are demonstrated in atomic ensembles, either at room temperature or in magneto-optical traps[13], and using trapped single atoms or ions[14,15]. However, there, the relatively weak light confinement achievable by tightly focusing a free-space laser beam, limited the achievable phase shift from a single atom to a few degrees[14]. Free-space, high finesse cavities were considered to increase the light-atom coupling[16,17], as well as their nanophotonic equivalents[18–20], enabling to drastically increase the reachable phase shift by single atoms, although at the cost of greater experimental complexity. In parallel, solid-state emitters have been considered a promising platform due to their ease of integration with nanophotonic structures[21], and significant phase shifts have been demonstrated in nanocavities[22]. There, the help of the Purcell effect enabled increasing the coupling

[1]Center for Hybrid Quantum Networks (Hy-Q), Niels Bohr Institute, University of Copenhagen, DK-2100 Copenhagen Ø, Denmark. [2]Lehrstuhl für Angewandte Festkörperphysik, Ruhr-Universität Bochum, Universitätsstraße 150, 44801 Bochum, Germany. ✉e-mail: lodahl@nbi.ku.dk; hanna.le-jeannic@cnrs.fr

efficiency to reduce the influence of decoherence channels. However, in a cavity, the quantum nonlinear response is limited to within the narrow cavity linewidth, which may limit the scalability of the approach. In nanophotonic waveguides, the Purcell enhancement is typically weaker yet the strong suppression of emission leakage entails that the photon-emitter coupling efficiency can be near unity[6], however the single-photon phase shift has been limited to only a few degrees because of the restricted coupling efficiency of molecules[23]. Among them, single QDs embedded in photonic waveguides can potentially reach very pronounced single-photon phase shifts, thanks to the high single-mode coupling efficiency[6] and nearly lifetime-limited emission lines[24].

In most experiments and protocols, the focus has been on measuring the intensity modification of a light field after interaction with the emitter[25–28], either in transmission ($I_t$) or in reflection ($I_r$). However, the direct measurement of the essential phase response of the non-linear interaction requires interferometric measurement of the optical response of the quantum emitter. Previous phase shift measurements include a direct measurement using Mach-Zehnder interferometry with a single atom in a focused beam[14], limited by the coupling efficiency. In contrast, using a heterodyne detection-like scheme, phase shifts induced by single organic molecules up to $0.017\pi$[23] were reconstructed, and more recently, even to $0.37\pi$[22] radians, the later demonstration being in a cavity-embedded scheming, reaching the strong coupling regime. The method established in that study demonstrated high resilience against thermal, mechanical, and optical disturbances. However, its implementation involved fitting Floquet theory for a single emitter interacting with two laser beams (and therefore to be considered an indirect measurement). This could pose challenges, particularly in experiments with QDs where multiple and broader transitions are situated in close proximity to each other. Further experiments show a $\simeq \pi$ phase shift, in the reflection of an atom coupled to a cavity[20,29].

In a waveguide, the transmission coefficient is defined as $t = \frac{\langle \hat{E}_{out} \rangle_{ss}}{\langle \hat{E}_{in} \rangle_{ss}}$, where $\hat{E}_{in}$ and $\hat{E}_{out}$ are the input and output field operators, respectively (see Fig. 1a), evaluated in the steady state (ss). The phase shift is expressed as its argument $\phi = \arg(t)$. In the case of a lifetime-limited quantum emitter of decay rate $\gamma$ and bidirectional (isotropic) interaction, the maximum single-photon phase shift achievable on resonance reaches $\pi/2$, in the limit where the light-matter coupling efficiency (the $\beta$-factor) reaches unity[4]. For $\beta \neq 1$, the phase shift is maximum for a light-emitter detuning of $\Delta = \pm\gamma \frac{\sqrt{1-\beta}}{2}$[23]

$$|\phi|_{max} = \tan^{-1}\left(\frac{\beta}{2\sqrt{1-\beta}}\right) \qquad (1)$$

(see Supplementary Information for the detailed calculation of the transmission coefficient). Recently, a photon-scattering reconstruction method was implemented to indirectly infer a phase shift of $0.22\pi$[30]. Here, we demonstrate the direct measurement of a single-photon phase shift induced from the interaction with a QD in a nanophotonic waveguide by implementing interferometric measurements.

## Results

### Experimental setup

The measurement setup, sketched in Fig. 1a, consists of a ~3 m long Mach-Zehnder interferometer built on top of a closed-cycle cryostat, where the nanophotonic chip is cooled down to 4 K. A continuous-wave laser is sent to one of the interferometer arms containing a GaAs photonic crystal waveguide with an InGaAs QD embedded inside (for more details on the sample fabrication, see ref. 31, see also Methods). After interaction with the QD, the signal is coupled out of the wave-guide chip and interfered with the reference arm (the local oscillator, LO). The achieved interferometer visibility is $\nu \approx 0.65$, mainly limited by the imperfect mode matching between the LO and the light

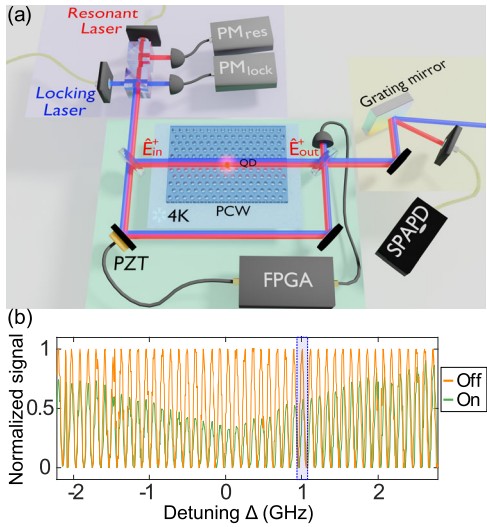

**Fig. 1 | Measurement of the phase shift. a** Experimental setup: a Mach-Zehnder interferometer is used to measure the phase shift caused by a single quantum dot (QD) in a photonic crystal waveguide(PCW) cooled to 4K. The interferometer is locked using a two-color scheme, where a far-detuned laser (blue) is used as a reference, and a feedback loop is implemented with a FPGA and a piezoelectric transducer (PZT). The low-power, resonant interference signal (red) is separated from the higher-power locking beam (blue) through a grating mirror. The filtered signal is then captured by a single-photon avalanche photodiode (SPAPD). $PM_{res}$ and $PM_{lock}$ are the two power meters used to stabilize the laser powers. **b** Evolution of the interference signal with detuning of the resonant laser (relative to the most pronounced QD transition) when the QD is tuned "on" (green). Same laser tuning range interference evolution when the QD is switched "off" (orange) through the application of an electric field across the QD (DC-Stark effect). A zoom-in of the blue area is presented in Fig. 2b.

out-coupled from the chip's gratings. The limited visibility only affects the signal-to-noise ratio of the measurement but suffices for resolving the narrow spectral features of the QD resonances. The resulting interference signal is then sent to a single-photon detector. To stabilize such a long interferometer, which is sensitive to sub-wavelength-scale vibrations, we apply a second laser, the locking laser, to measure and implement fast feedback corrections on the optical path (see the Method Section for more details). Finally, the locking laser is filtered from the signal using a grating filter setup.

### Measurement of a phase shift across the resonance

To probe the phase shift, the frequency of the resonant laser is swept across the QD resonance to measure the resulting interference signal, while the locking laser frequency stays fixed. We tune the resonance frequency of the QD with a voltage applied across the sample by virtue of the DC-Stark effect[31], allowing us to compare the on- and off-resonance cases, respectively (See Fig. 1b), and determine directly and accurately the phase shift induced by the QD (see the Method Section and the Supplemental materials for more details). Figure 2a, b presents two examples of signals at different laser detunings. Away from resonance (Fig. 2a), no significant intensity and phase change is observed, meaning the change of the electric field itself does not affect the laser transmission, while near resonance (Fig. 2b), the fringe contrast and phase changes when the QD is set to be resonant with the laser field. Through a single measurement, we can thus infer both the phase and intensity changes experienced by the light field due to the interaction with the QD. The results are presented in Fig. 2, where the phase (c) and intensity (d) spectra of the two dipole transitions of the QD neutral exciton, labeled (1) and (2), are displayed. We fit the phase and intensity data of both dipoles simultaneously (See Supplementary Information), and infer the maximal phase shifts to be $\phi_{max,1} = (-0.06 \pm 0.03)\pi$ and $\phi_{max,2} = (-0.19 \pm 0.03)\pi$

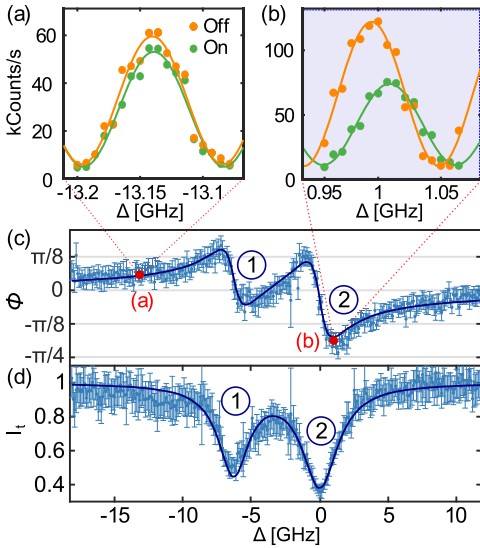

**Fig. 2 | Spectrum of the phase shift. a, b** Direct interferometric data with the emitter tuned "on" (green) and "off" (orange) resonance using the external electric field, for two different laser-emitter detunings (integration time of 100 ms per point). The measurement points are plotted along with corresponding sinusoidal fits (solid line). The data in **b** correspond to the detuning area marked in blue in Fig. 1b. **c, d** Extracted respective phase shift and transmission for the two dipoles, labeled 1 and 2. The solid lines correspond to the fit of the data to the theory. Errors bars represent the standard deviation of the signal.

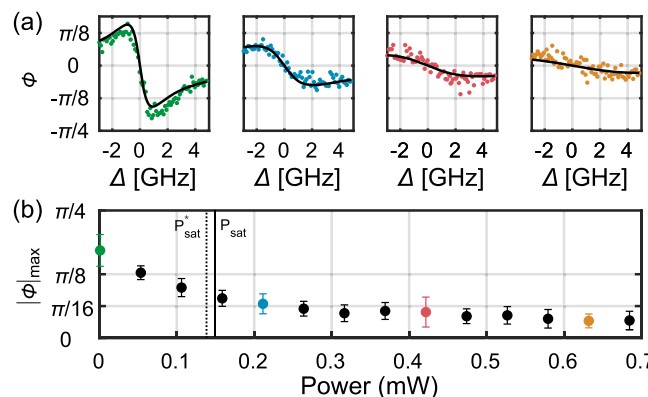

**Fig. 3 | Power response of the phase shift. a** Measurements of the phase response of the QD versus detuning and for different excitation powers. The solid lines are the fit to the theory of the overall data set. **b** Maximum measured experimental phase shift as a function of input power measured at $PM_{res}$, see Fig. 1a. The colored points correspond to the data shown in **a**. The full horizontal line indicates the calculated saturation power of the transition $P_{sat} = 0.15$ mW. The dashed line indicates the saturating power $P_{sat}^* = 0.14$ mW of the maximal phase shift (See Supplementary Information). Errors bars represent the standard deviation of the signal.

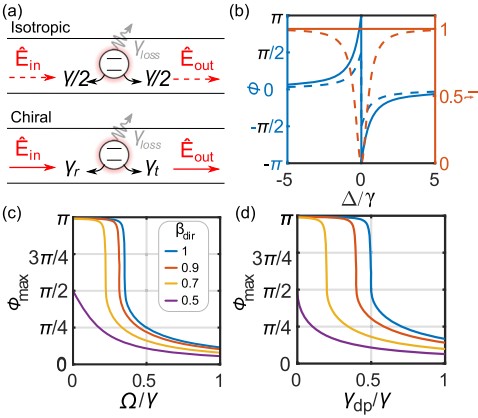

**Fig. 4 | Isotropic and chiral configurations. a** shows the scattering configuration for an isotropic and a chirally coupled system. In the latter, the reflection and transmission decay rates ($\gamma_r$ and $\gamma_t$ respectively) differ. Correspondingly, **b** shows the phase shift and transmission intensity for the isotropic (dashed line) and chiral coupling (full line). **c** Maximal phase shift $\phi_{max}$ as a function of the driving Rabi frequency $\Omega$ for different directional coupling efficiencies $\beta_{dir} = [1(blue), 0.9(red)$ $0.7(yellow)0.5(purple)]$. $\beta_{dir} = 0.5$ corresponds to the case of an isotropic waveguide with $\beta = 1$. **d** $\phi_{max}$ as a function of the pure dephasing rate $\gamma_{dp}$ for a series of coupling efficiency $\beta_{dir} = \{1$ (blue), 0.9 (red) 0.7 (yellow) 0.5 (purple)\}.

radians, respectively. We have thus presented a method of directly measuring the total transmission response across the resonance of an emitter in a waveguide. The phase shift is about thirty times larger than a previous direct measurement using Mach-Zehnder interferometry[14], yet is limited by residual broadening of the QD emission line. The method itself is only limited to the signal intensity and, similarly, the integration time per point.

## Saturation measurement

Next, we examine the saturation of the phase shift in order to investigate its nonlinear response to changes in the incoming laser power. We consider dipole transition (2). In Fig. 3a, we show several spectra taken at different laser power levels and the corresponding fitting of the full saturation behavior (see Supplementary Information), which is fully consistent with the data presented. For each power level, we determine the maximum experimentally observed phase shift and investigate the nonlinear behavior as the QD saturates, see Fig. 3b. By using the experimental parameters extracted previously, we estimate that the saturation happens at a mean photon flux of $n_c \sim 0.39$ photons interacting with the QD during its lifetime (See ref. 30 and Supplementary Information), well below the single-photon level. This should enable observing a differential phase shift between single and two-photon components (also often called "nonlinear" phase shift), such as measured in ref. 19, essential for the implementation of controlled quantum operations.

## Discussion

The experimentally extracted phase shifts are limited by the coupling efficiency and decoherence of the QD, and future experiments on fully lifetime-limited QD transitions[24] should allow observing a phase shift approaching $\pi/2$. Going beyond this would even be possible in the setting of chiral quantum optics[32] where directional coupling entails that the reflective loss channel can be strongly suppressed Fig. 4a schematically illustrates the isotropic and chiral cases, respectively. In the ideal chiral case, the maximum possible phase shift of $\pi$ can be realized, the ultimate goal for quantum phase gates[1,33,34]. In contrast,

the transmitted intensity would be unchanged at resonance, see Fig. 4b, i.e., no photons are lost and the scattering is thereby deterministic in transmission. Such a single-photon response, however, would be undetectable in intensity measurements and, therefore, require the interferometric method demonstrated here. It is interesting to further exploit the unusual behavior of the phase response in the chiral geometry. When the input light intensity is increased, a very abrupt phase response is predicted (see Fig. 4c), unlike in the symmetric configuration. Indeed, towards saturation, the transmission coefficient at resonance (which is real) changes from a negative value to a positive value, resulting in a sudden shift in the phase from $\pi$ to 0. This may find applications as an all-optical phase-switch[20,23,35]. Similarly, a sharp transition can be found while varying the dephasing rate (see Fig. 4d), which means it may be applicable as an ultra-sensitive probe of environmental decoherence processes of the QD. Finally, we rediscover that the case of ideal directionality is equivalent to an ideal

emitter in an isotropic waveguide when the efficiency decreases by half due to saturation ($\Omega \geq \frac{\gamma}{2\sqrt{2}}$), dephasing ($\gamma_{dp} \geq \gamma/2$), or coupling inefficiency ($\beta_{dir} \leq 1/2$).

In summary, we have developed an interferometric method for measuring the nonlinear phase shift of light caused by a single quantum emitter and measured an unprecedented phase response in a waveguide. These results may open up a wide range of applications on how to realize deterministic quantum phase gates in photonic circuits[33,36] as a basis for quantum non-demolition measurements[17,18] or deterministic generation of optical Schrödinger cat states[37], when combined with accurate spin control[34,38,39]. This work holds promises for on-chip photonic quantum processing, in particular combined with the recent achievement on the integration[40] and coherent coupling[41] of multiple quantum dots in waveguides. Additionally, the quantum emitter phase shift may be applied as the quantum nonlinear operation required in quantum optical neural networks[3], where even moderate nonlinear phase shifts have been shown to suffice for improving the implementation of Bell-state detectors[11,12]. Finally, chiral light-matter interaction promises to improve the phase response even further, although the combination of a high $\beta$-factor and high directionality has not yet explicitly been demonstrated in a waveguide. In such a configuration, interferometric measurements are required to detect the single-photon-scattering processes, and the complex phase response acquired by optical pulses constitutes an interesting future direction of research that may also shed new light on applications of the emitter nonlinearity.

## Methods

Our QD was embedded in a photonic crystal waveguide with a radius of 70 nm and a lattice constant of 250 nm. The sample configuration closely resembled the layout utilized in ref. 24. The bandgap was positioned ~0.5 THz away from the emission wavelength resulting in only weak Purcell enhancement while still maintaining a high $\beta$-factor[24]. Light was coupled to and from the chip through shallow-etched grating couplers, where efficiencies of >25%, are typically reached while grating back reflections are strongly suppressed[42].

The QD was tuned in and out of resonance through DC-stark shift tuning using a voltage field. The "off" state (corresponding to an applied voltage of 0.8 V) was checked to be away from any optically active transition of the dot. The "on" voltage was set to 1.24 V. The linewidths of the two quantum dot dipole transitions were fitted to be $1.95 \pm 0.05$ GHz and $1.45 \pm 0.05$ GHz wide.

The interferometer is locked by having a piezoelectric transducer-mounted mirror to compensate for any change in phase not originating from the quantum emitter (see Supplemental Information for more details on the experimental setup). The feedback is performed by using an FPGA (Field Programmable Array, Red Pitaya) programmed to act like a lock-in amplifier followed by a proportional-integral-derivative controller[43].

The locking laser is blue-detuned by 7.5 nm from the QD transition at 941 nm to avoid any interaction with the emitter and at a much higher power than the few-photon resonant laser. This wavelength was chosen to stay away from the QD transition while keeping a good transmission in the sample (away from the bandgap). We saw no difference compared to the use of a red-detuned laser. The response frequency of the mirror and piezoelectric system is limited to 4 kHz. The lock-in modulation signal was chosen to be driven at 3.1 kHz.

## Data availability

Data sets generated during the current study are available from the corresponding author upon reasonable request. Source data are provided in this paper.

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

## Acknowledgements
We thank Vasiliki Angelopoulou for her help at the early stages of the experiment. We acknowledge funding from the Danish National Research Foundation (Center of Excellence "Hy-Q," Grant no. DNRF139) and from the European Union's Horizon 2020 research and innovation programs under Grant Agreements no. 824140 (TOCHA, H2020-FET-PROACT-01-2018). This project has also received funding from BMBF 16KIQS009.

## Author contributions
M.J.R.S., H.L.J., and A.T. carried out the experiment and analyzed the data. M.J.R.S. and H.L.J. wrote the manuscript with support from all authors. Y.W., S.S., A.D.W., and A.L. fabricated the sample with the help of L.M. H.L.J., N.R., and P.L. supervised the project.

## Competing interests
Peter Lodahl is the founder of the start-up company Sparrow Quantum.
