## [Peer Review File · Nature Communications]

Direct observation of a few-photon phase shift induced by a single quantum emitter in a waveguideEditorial note: This manuscript has been previously reviewed at another journal that is not operating a transparent peer review scheme. This document only contains reviewer comments and rebuttal letters for versions considered at *Nature Communications*.

REVIEWER COMMENTS

Reviewer #1 (Remarks to the Author):

I am satisfied with the author's response to my previous comments. I think the experimental results here further demonstrates the potential of their waveguide quantum emitter interface towards on-chip quantum information, particularly considering the complexity of introducing it into a technically demanding experimental apparatus. I find that the manuscript is appropriate for Nature Communications as is, and am happy to recommend publication.

Reviewer #2 (Remarks to the Author):

The authors have thoroughly addressed all my comments. Consequently, I now recommend publication of the paper.

Reviewer #3 (Remarks to the Author):

I have reviewed the responses to my previous comments and thank them for their efforts. As I wrote before, I acknowledge the high technical level of their results. In particular, I now understand that the present result has been achieved by high level techniques for the interferometric experiments. Unfortunately, however, I am still not fully convinced that the implications of this result over the previous results are clearly presented in the revised manuscript. I list below my concerns that make me hesitate to make a decision.

1) If I understand correctly, the application for quantum information and networking in general requires the nonlinear phase shift over π . Since the theoretically achievable shift is smaller than $\pi/2$ in the QD waveguide system, there is a significant drawback compared to

the QD cavity systems where Pi-shift has already been achieved. Do the authors think that it is necessary to finally implement chiral optics in their QD waveguide system, or are there specific applications where the phase shift is smaller than $\text{Pi}/2$? I recommend them to clarify this issue in the manuscript. In their last reply, they mentioned that the implementation of chiral optics to overcome this barrier seems rather difficult, which makes me wonder about the goal of this result.

2) In the last review, I commented, "In this regard, it is not clear to me whether the result presented in Fig. 3 does have an impact against the previous reports in terms of nonlinearity", where I meant that the authors should compare this saturation behavior with the nonlinear shift of the single-photon level in the previous results, such as the saturation photon number, the saturation value of the shift, and the shape of the saturation curve, etc.

3) As I pointed out earlier, although the title says "nonlinear phase shift", most of the present data is about linear shifts. If I understand correctly, the 0.19 Pi is the linear shift. I recommend that the authors should distinguish them, for example in the abstract, and clarify the value of the exact nonlinear shift in their experiment. My guess is that the nonlinear shift in Fig. 3 is smaller than 0.19 Pi . For the same reason, the saturation photon number is an important quantity. I recommend to show the saturation power in Fig. 3. Although they derived the photon number from fitting, the saturation power can be obtained independently from the input power in Fig. 3. This check would be worth showing to strengthen their result.

4) The authors did not provide the response to my following comment. "In introduction, they mentioned that most previous works employ intensity measurements, and emphasized the importance of the interferometric measurements. However, there are many works of interferometric measurements in quantum optics with atoms and molecules. The phrase would be misleading." I still think that they should describe about previous interferometric measurements in quantum optics, and should show the technical difference.

5) In response to my previous comment, they showed the value of the coupling efficiency.

But they wrote that they did not measure it for their device. Since this value is important, they should present how they estimated the efficiency value exactly.

Reviewer #3 (Remarks to the Author):

I have reviewed the responses to my previous comments and thank them for their efforts. As I wrote before, I acknowledge the high technical level of their results. In particular, I now understand that the present result has been achieved by high level techniques for the interferometric experiments. Unfortunately, however, I am still not fully convinced that the implications of this result over the previous results are clearly presented in the revised manuscript. I list below my concerns that make me hesitate to make a decision.

We thank the reviewer for their previous comments and for acknowledging the high technical quality of our results. We hope that our present reply and the changes made to the manuscript adequately address their concerns.

1) If I understand correctly, the application for quantum information and networking in general requires the nonlinear phase shift over π . Since the theoretically achievable shift is smaller than $\pi/2$ in the QD waveguide system, there is a significant drawback compared to the QD cavity systems where π -shift has already been achieved. Do the authors think that it is necessary to finally implement chiral optics in their QD waveguide system, or are there specific applications where the phase shift is smaller than $\pi/2$? I recommend them to clarify this issue in the manuscript. In their last reply, they mentioned that the implementation of chiral optics to overcome this barrier seems rather difficult, which makes me wonder about the goal of this result.

We appreciate the reviewer's insightful question. It is true that π phase shifts have been achieved in cavity systems. However, waveguides, while relatively new, offer the advantage of operating over a much broader wavelength range, though they are more sensitive to decoherence in the absence of the Purcell effect. Additionally, in cQED, large phase shifts often require strong coupling (as demonstrated in the seminal work by Volz et al.[1]). In contrast, in wQED, we operate in the weak coupling regime.

Regarding the phase shift, achieving a π phase shift is ideal, but there are applications where even moderate phase shifts, well below $\pi/2$ are sufficient. For instance, protocols like Bell measurements can be implemented with phase shifts up to $\pi/4$ (as discussed by Ewaniuk et al., [2].)

As for integrating chiral optics into our QD waveguide system, we think that it is a promising direction. Nonetheless, developing protocols that are more tolerant to the inherent decoherence in quantum systems is also crucial. This dual approach can help maximize the potential of emitters in waveguides in practical quantum information applications.

We had previously written a perspective on this in the end of the article : “Additionally, the quantum emitter phase shift may be applied as the quantum nonlinear operation required in quantum optical neural networks [1], where even moderate nonlinear phase shifts have been shown to suffice for improving the implementation of Bell-state detectors[3,2].”

[1] olz, J., Scheucher, et al. Nonlinear π phase shift for single fibre-guided photons interacting with a single resonator-enhanced atom. Nature Photon 8, 965–970 (2014).

- [2] J. Ewaniuk, et al., Imperfect Quantum Photonic Neural Networks. Adv Quantum Technol. 2023, 6, 2200125.
- [3] A. Pick, et al., Boosting Photonic Quantum Computation with Moderate Nonlinearity, Phys. Rev. Applied 15, 054054 (2021)
- [4]: Steinbrecher, et al. Quantum optical neural networks. npj Quantum Inf 5, 60 (2019)

Changes in the Revised Manuscript:

We have added the following sentences in the introduction to precise this :

“In the waveguide geometry, a narrow-band single-photon wavepacket is deterministically reflected upon resonant interaction with a highly coherent two-level quantum emitter, while two-photon wavepackets are partly transmitted due to the saturation of the emitter [7,8], allowing for realizing deterministic quantum operations such as photon sorters [9,10].

In contrast to optimal π phase shift operations, even moderate non-linear interactions have been proposed as a way to boost measurement-based quantum computing [11] and for the implementation of quantum neural networks [3,12]. ”

2) In the last review, I commented, "In this regard, it is not clear to me whether the result presented in Fig. 3 does have an impact against the previous reports in terms of nonlinearity", where I meant that the authors should compare this saturation behavior with the nonlinear shift of the single-photon level in the previous results, such as the saturation photon number, the saturation value of the shift, and the shape of the saturation curve, etc.

We thank the reviewer for these suggestions.

The critical photon flux for a waveguide coupled emitter may be calculated according to $n_c = (1 + 2\beta\gamma_{dp}/\gamma)/4\beta^2$ [1], which, in the limit of Fourier transform-limited emitters and perfect waveguide-coupling, reduces to $n_c = 1/4$. From the parameters extracted from the fitting of the saturation, we find a critical photon flux $n_c = 0.39[0.25,0.73]$. We can compare to this to previous measurements:

1. Quantum dots in waveguides: (weak coupling regime) :

Javadi et al., Nature Communications 6, 8655 (2015)	$n_c \sim 0.81$
Thyrrestrup et al., Nano Lett., 18, 3, 1801–1806 (2018)	$n_c \sim 1.6$
Le Jeannic et al., Phys. Rev. Lett. 126, 023603 (2021)	$n_c \sim 0.33$

2. Organic molecule in free space (weak coupling regime):

M. Pototschnig et al., Phys. Rev. Lett. 107 (2011)	$n_c \sim 25$ ($\eta = \beta = 0.1$)
--	--

3. Quantum dot in cavity: (strong coupling regime) Fushman et al., Science 320, 5877 pp. 769-772 (2008)	$n_c \sim 0.4 - 0.6$
--	----------------------

4. Atom coupled to a resonator:(strong coupling regime) Volz et al., Nature Photonics volume 8, p. 965–970 (2014)	$n_c = \gamma^2/(2g^2) \approx 0.02$
--	--------------------------------------

5. Organic molecule in cavity: (strong coupling regime)

Changes in the Revised Manuscript:

We have added the comparison to other systems in the supplementary material as:

Similarly, we can estimate the critical photon flux from the fitted parameters as: $n_c = \frac{1+2\beta\gamma_{dp}/\gamma}{4\beta^2} \sim 0.39[0.25, 0.73]$ [4,5]. This is comparable to previously measured values for solid state emitters: quantum dots: $n_c \sim [0.81[5], 0.33[4], 1.6[6]]$ in weak coupling and $n_c \sim 0.4 - 0.6$ [7] in strong coupling; or in organic molecules: $n_c \sim 25$ in weak coupling [8] and $n_c \sim 0.44$ [9] in strong coupling. An atom strongly coupled to a resonator has demonstrated even lower values, with $n_c \sim 0.02$ [10]."

We also corrected an error in the critical photon number calculation, which should be **0.39, not 0.33.**

3) As I pointed out earlier, although the title says "nonlinear phase shift", most of the present data is about linear shifts. If I understand correctly, the 0.19π is the linear shift. I recommend that the authors should distinguish them, for example in the abstract, and clarify the value of the exact nonlinear shift in their experiment. My guess is that the nonlinear shift in Fig. 3 is smaller than 0.19π . For the same reason, the saturation photon number is an important quantity. I recommend to show the saturation power in Fig. 3. Although they derived the photon number from fitting, the saturation power can be obtained independently from the input power in Fig. 3. This check would be worth showing to strengthen their result.

We apologize for not fully understanding the reviewer's point earlier. In our terminology, "nonlinear phase shift" refers to the total phase shift experienced by the beam as a function of power (mean photon number). If we now understand correctly, the reviewer is referring to the nonlinear phase shift as the difference between the single-photon phase shift and the two-photon phase shift, as for example defined in Volz et al.[10]

Indeed, the 0.19π phase shift observed in our experiments can be considered close to the single-photon limit, denoted in Ref. [10] as ϕ_1 . To determine the two-photon phase shift (ϕ_2) and thus the nonlinear phase shift ($\phi_2 - 2\phi_1$), a two-photon tomography experiment would be required, similar to the approach taken by Volz et al. [10].

Changes in the Revised Manuscript:

We thank the reviewer for bringing this important distinction to our attention and have clarified this in our manuscript to avoid any confusion.

1. We have changed the title by "Direct observation of **a few-photon phase shift** induced by a single quantum emitter in a waveguide"
2. Additionally we have changed in the main text the following sentence in the abstract: **"The process is nonlinear in power, the saturation at the single-photon level** and compatible with scalable photonic integrated circuitry."
3. We also have added the sentence in the main text **"This should enable the observation of a differential phase shift between single and two-photon components (also often called**

"nonlinear" phase shift), such as measured in Ref. [19], which is essential for the implementation of controlled quantum operations."

3/b Regarding the saturation power, we understand the importance of this quantity. The issue with defining a saturation power using only Fig. 3b is that each $|\phi_{max}|$ is taken at a different detuning, thus not following a classical saturation curve at fixed detuning.

The "real" saturation power in the classical sense is defined at a fixed detuning. For example, at resonance $\Delta = 0$, the saturation parameter reads $S = \frac{4\Omega^2}{\gamma\gamma_2}$ ($\rho_{ee} = \frac{S}{2(S+1)} = 1/4$

at $S=1$). Using the fitted parameters, this gives a saturation power at resonance:

$$P_{sat} = \frac{\gamma(\gamma/2 + \gamma_{dp})}{4\eta} \approx 0.15mW.$$

However, the saturation power cannot easily be extracted at resonance since the phase shift quickly goes to zero with power around the resonance (see Figure below).

To evaluate a saturation power (denoted P_{sat}^*) of the maximal phase shift without assuming any parameters, we fit the data using a general decaying model $|\phi_{max}| = Ae^{-P/P_{sat}^*} + B$. This yields $P_{sat}^* = 0.14$ [0.11,0.16] mW, corresponding to a maximal phase shift of 0.27 rad.

Changes in the Revised Manuscript:

We thank the reviewer for this point and have now indicated P_{sat} and P_{sat}^* on Fig. 3b. This calculation has been added to the supplementary material.

"Those values are in good agreement with the data of the two dipoles in Figure 2.

Since the exact Rabi frequency is unknown, we define the mapping constant η to the power as $\Omega = \sqrt{\eta P}$. At resonance, the saturation parameter is given by $S = \frac{4\Omega^2}{\gamma\gamma_2}$ (where $\rho_{ee} = \frac{S}{2(S+1)} = 1/4$ at $S=1$). Using the fitted parameters, this yields: $P_{sat} = \frac{\gamma(\gamma/2 + \gamma_{dp})}{4\eta} \approx 0.15mW$ (indicated by the solid line in Fig. 3(b)).

Similarly, we can estimate the critical photon flux from the fitted parameters as: $n_c = \frac{1+2\beta\gamma_{dp}/\gamma}{4\beta^2} \sim 0.39$ [0.25,0.73] [4,5]. This is comparable to previously measured values for solid state emitters: quantum dots: $n_c \sim [0.81$ [5], 0.33 [4], 1.6 [6]] in weak coupling and $n_c \sim 0.4 - 0.6$ [7] in strong coupling; or in organic molecules: $n_c \sim 25$ in weak coupling [8] and $n_c \sim 0.44$ [9]

in strong coupling. An atom strongly coupled to a resonator has demonstrated even lower values, with $n_c \sim 0.02$ [10].

At each power we also perform an independent fit with free parameters, to extract accurately the maximal, experimentally measured phase shift from the data. These data points $|\phi_{max}|$ are displayed in Fig.3(b). **By fitting with a general decay model $|\phi_{max}| = Ae^{-P/P_{sat}^*} + B$, we find $P_{sat}^* = 0.14$ [0.11,0.16] mW. This P_{sat}^* represents the saturation power of the maximum phase shift and is indicated by the dotted line in Fig. 3(b)."**

Fig3: (a) Measurements of the phase response of the QD versus detuning and for different excitation powers. The solid lines are the fit to the theory of the overall data set. (b) Maximum measured experimental phase shift as a function of input power (measured at PM_{res} , see Fig1.(a)). The colored points correspond to the data shown in (a). **The solid horizontal line represents the calculated saturation power of the transition, $P_{sat} = 0.15$ mW. The dotted line indicates the saturation power of the maximal phase shift, $P_{sat}^* = 0.14$ mW (see Supplementary Information).**

4) The authors did not provide the response to my following comment. "In introduction, they mentioned that most previous works employ intensity measurements, and emphasized the importance of the interferometric measurements. However, there are many works of interferometric measurements in quantum optics with atoms and molecules. The phrase would be misleading." I still think that they should describe about previous interferometric measurements in quantum optics, and should show the technical difference

We apologize to the reviewer in missing their point. To clarify that previous experiments have indeed been performed before we have now moved the citations of previous interferometric measurements to the introduction.

Changes in the Revised Manuscript:

The introduction now reads:

*“In most experiments and protocols, the focus has been on measuring the intensity modification of a light field after interaction with the emitter [25-28], either in transmission (I_t) or in reflection (I_r). However, the direct measurement of the essential phase response of the nonlinear interaction requires interferometric measurement of the optical response of the quantum emitter. **Previous phase shift measurements include a direct measurement using Mach-Zehnder interferometry with a single atom in a focused beam [14], limited by the coupling efficiency. In contrast, using a heterodyne detection-like scheme, phase shifts induced by single organic molecules up to 0.017π [23] were reconstructed, and more recently even to 0.37π [22] radians, the later demonstration being in a cavity-embedded scheme, reaching the strong coupling regime. The method established in that study demonstrated high resilience against thermal, mechanical, and optical disturbances. However, its implementation involved fitting Floquet theory for a single emitter interacting with two laser beams (and therefore to be considered an indirect measurement). This could pose challenges, particularly in experiments with QDs where multiple and broader transitions are situated in close proximity to each other.***

Further experiments show a $\approx \pi$ phase shift, in the reflection of an atom coupled to a cavity [20,29].”

The previous paragraph later on in the manuscript:

“In contrast, using heterodyne interferometry, phase shifts induced by single organic molecules up to 0.017π [25] were reconstructed, and more recently even to 0.37π [24] radians, the later demonstration being cavity-embedded. The method established in that study demonstrated high resilience against thermal, mechanical, and optical disturbances. However, its implementation involved fitting Floquet theory for a single emitter interacting with two laser beams (and therefore to be considered an indirect measurement). This could pose challenges, particularly in experiments with QDs where multiple and broader transitions are situated in close proximity to each other. Further experiments show a $\approx \pi$ phase shift, in the reflection of an atom coupled to a cavity [22,29]”

now reads:

“We have thus presented a method of directly measuring the total transmission response across the resonance of an emitter in a waveguide. The phase shift is about thirty times larger than a previous direct measurement using Mach-Zehnder interferometry [14], yet limited by residual broadening of the QD emission line. The method itself is only limited to the signal intensity and similarly the integration time per point.”

5) In response to my previous comment, they showed the* value of the coupling efficiency. But they wrote that they did not measure it for their device. Since this value is important, they should present how they estimated the efficiency value exactly.

The coupling efficiency was determined by fitting the data from Fig 3, i.e. the phase shift spectra serie with power. This inherently introduces significant error bars, due to the number of parameters in the system. More complex experiments are required for precise extraction, as highlighted by studies such as [1-3]. Extracting precisely the beta-factor in fact remains an active area of research within the community (see recent work from Scarpelli et al. [4]). Consequently, we emphasize the importance of having direct methods to measure the phase shift without relying on system parameter assumptions. This was the primary focus of our work.

[1] :H. Thyrestrup, L. Sapienza and P. Lodahl, Appl. Phys. Lett. 96, 231106 (2010)

[2] M. Arcari et al., Phys. Rev. Lett. 113, 093603, (2014)

[3] H. Le Jeannic et al., Phys. Rev. Lett. 126, 023603 (2021)

[4] L. Scarpelli, et al., Phys. Rev. B 100, 035311 (2019)

Changes in the Revised Manuscript:

We agree with the reviewer that our previous explanation lacked precision. We have now added more details to clarify our method:

We fit **all the transmission spectra series with power presented in Fig. 3 simultaneously using a nonlinear least-squares regression based on our model:**

$$\phi(\Delta, \Omega) = \arg\left(1 - \frac{\beta\gamma(\gamma_2 + i\Delta)}{2(\Delta^2 + \gamma_2^2 + 4\Omega^2\gamma_2/\gamma)}\right)$$

Since we do not know the exact Rabi frequency we define the mapping efficiency constant η to the power as $\Omega = \sqrt{\eta P}$

We obtain the **following single** set of parameters:

β	0.99~[0.57,1]
γ [ns^{-1}]	12.6~[7.7, 17.4]
γ_{dp} [ns^{-1}]	3.4 [0,7.4]
ϕ_0 [rad]	-0.26~ [-0.31,-0.2]
η [$rad (mW s)^{-1}$]	5 ~ [2.3, 7.7]

Direct observation of a few-photon phase shift induced by a single quantum emitter in a waveguide

Mathias J.R. Staunstrup,¹ Alexey Tiranov,¹ Ying Wang,¹ Sven Scholz,² Andreas D. Wieck,² Arne Ludwig,² Leonardo Midolo,¹ Nir Rotenberg,¹ Peter Lodahl,¹ and Hanna Le Jeannie¹

¹*Center for Hybrid Quantum Networks (Hy-Q), Niels Bohr Institute, University of Copenhagen, DK-2100 Copenhagen Ø, Denmark*

²*Lehrstuhl für Angewandte Festkörperphysik, Ruhr-Universität Bochum, Universitätsstraße 150, 44801 Bochum, Germany*

(Dated: June 19, 2024)

Realizing a sensitive photon-number-dependent phase shift on a light beam is required both in classical and quantum photonics. It may lead to new applications for classical and quantum photonics machine learning or pave the way for realizing photon-photon gate operations. Non-linear phase-shifts require efficient light-matter interaction, and recently quantum dots coupled to nanophotonic devices have enabled near-deterministic single-photon coupling. We experimentally realize an optical phase shift of $0.19\pi \pm 0.03$ radians (≈ 34 degrees) using a weak coherent state interacting with a single quantum dot in a planar nanophotonic waveguide. The phase shift is probed by interferometric measurements of the light scattered from the quantum dot in the waveguide. **The process is nonlinear in power, the saturation** at the single-photon level and compatible with scalable photonic integrated circuitry. The work may open new prospects for realizing high-efficiency optical switching or be applied for proof-of-concept quantum machine learning or quantum simulation demonstrations.

Optical nonlinearities are at the core of many modern applications in photonics. If sensitive at the level of single light quanta, they may be applied to realize fundamental quantum gate operations for photonic quantum computing or advanced quantum network implementations^{1,2}. The nanophotonics platform could potentially be scaled up to realize large-scale nonlinear quantum photonic circuits, as required, e.g., in quantum neural networks³. Strong optical nonlinearities can be achieved using single emitters such as molecules or quantum dots (QDs) embedded in photonic waveguides or cavities^{4,5} due to the tight confinement of light to reach light-matter coupling efficiencies near unity⁶. In the waveguide geometry, a narrow-band single-photon wavepacket is deterministically reflected upon resonant interaction with a highly coherent two-level quantum emitter, while two-photon wavepackets are partly transmitted due to the saturation of the emitter^{7,8}, allowing for realizing deterministic quantum operations such as photon sorters^{9,10}. **In contrast to optimal $\pi(\pi/2)$ phase shift operations, even moderate non-linear interactions have been proposed as a way to boost measurement-based quantum computing¹¹ and for the implementation of quantum neural networks^{3,12}**

Emitter-induced phase shifts demonstrated in atomic ensembles, either at room temperature or in magneto-optical traps,¹³ and using trapped single atoms or ions^{14,15}. However, there, the relatively weak light confinement achievable by tightly focusing a free-space laser beam, limited the achievable phase shift from a single atom to a few degrees¹⁴. Free-space, high finesse cavities were considered to increase the light-atom coupling^{16,17}, as well as their nanophotonic equivalents^{18–20}, enabling to drastically increase the reachable phase shift by single atoms, although at the cost of greater experimental complexity. In parallel, solid-state emitters have been considered a promising platform due to their ease of in-

tegration with nanophotonic structures²¹ and significant phase shifts have been demonstrated in nanocavities²². There, the help of the Purcell effect enabled increasing the coupling efficiency to reduce the influence of decoherence channels. However, in a cavity the quantum nonlinear response is limited to within the narrow cavity linewidth, which may limit the scalability of the approach. In nanophotonic waveguides, the Purcell enhancement is typically weaker yet the strong suppression of emission leakage entails that the photon-emitter coupling efficiency can be near unity⁶, however the single-photon phase shift has been limited to only a few degrees because of the restricted coupling efficiency of molecules²³. Among them, single QDs embedded in photonic waveguides can potentially reach very pronounced single-photon phase shifts, thanks to the high single-mode coupling efficiency⁶ and nearly lifetime-limited emission lines²⁴.

In most experiments and protocols, the focus has been on measuring the intensity modification of a light field after interaction with the emitter^{25–28}, either in transmission (I_t) or in reflection (I_r). **However, the direct measurement of the essential phase response of the non-linear interaction requires interferometric measurement of the optical response of the quantum emitter. Previous phase shift measurements include a direct measurement using Mach-Zehnder interferometry with a single atom in a focused beam¹⁴, limited by the coupling efficiency. In contrast, using a heterodyne detection-like scheme, phase shifts induced by single organic molecules up to 0.017π ²³ were reconstructed, and more recently even to 0.37π ²² radians, the later demonstration being in a cavity-embedded scheming, reaching the strong coupling regime. The method established in that study demonstrated high resilience against thermal, mechanical, and optical disturbances. However, its implementa-**

tion involved fitting Floquet theory for a single emitter interacting with two laser beams (and therefore to be considered an indirect measurement). This could pose challenges, particularly in experiments with QDs where multiple and broader transitions are situated in close proximity to each other. Further experiments show a $\simeq \pi$ phase shift, in the reflection of an atom coupled to a cavity^{20,29}.

In a waveguide, the transmission coefficient is defined as $t = \frac{\langle \hat{\mathbf{E}}_{\text{out}} \rangle_{\text{ss}}}{\langle \hat{\mathbf{E}}_{\text{in}} \rangle_{\text{ss}}}$, where $\hat{\mathbf{E}}_{\text{in}}$ and $\hat{\mathbf{E}}_{\text{out}}$ are the input and output field operators, respectively (see Fig. 1(a)), evaluated in the steady state (ss). The phase shift is expressed as its argument $\phi = \arg(t)$. In the case of a lifetime-limited quantum emitter of decay rate γ and bidirectional (isotropic) interaction, the maximum single-photon phase shift achievable on resonance reaches $\pi/2$, in the limit where the light-matter coupling efficiency (the β -factor) reaches unity⁴. For $\beta \neq 1$, the phase shift is maximum for a light-emitter detuning of $\Delta = \pm \gamma \frac{\sqrt{1-\beta}}{2}$ ²³

$$|\phi|_{\text{max}} = \tan^{-1} \left(\frac{\beta}{2\sqrt{1-\beta}} \right) \quad (1)$$

(see Supplementary Information for the detailed calculation of the transmission coefficient). Recently, a photon-scattering reconstruction method was implemented to indirectly infer a phase shift of 0.22π ³⁰. Here, we demonstrate the direct measurement of a single-photon phase shift induced from the interaction with a QD in a nanophotonic waveguide by implementing interferometric measurements.

The measurement setup, sketched in Fig. 1(a), consists of an approximately 3m long Mach-Zehnder interferometer built on top of a closed-cycle cryostat, where the nanophotonic chip is cooled down to 4K. A continuous-wave laser is sent to one of the interferometer arms containing a GaAs photonic crystal waveguide with an InGaAs QD embedded inside (for more details on the sample fabrication, see³¹, see also Methods). After interaction with the QD, the signal is coupled out of the waveguide chip and interfered with the reference arm (the local oscillator, LO). The achieved interferometer visibility is $v \approx 0.65$, mainly limited by the imperfect mode matching between the LO and the light out-coupled from the chip's gratings. The limited visibility only affects the signal-to-noise ratio of the measurement but suffices for resolving the narrow spectral features of the QD resonances. The resulting interference signal is then sent to a single-photon detector. To stabilize such a long interferometer, which is sensitive to sub-wavelength-scale vibrations, we apply a second laser, the locking laser, to measure and implement fast feedback corrections on the optical path (see the Method Section for more details). Finally, the locking laser is filtered from the signal using a grating filter setup.

To probe the phase shift, the frequency of the resonant laser is swept across the QD resonance to measure

FIG. 1. (a) Experimental setup: a Mach-Zehnder interferometer is used to measure the phase shift caused by a single quantum dot (QD) in a photonic crystal waveguide (PCW) cooled to 4 K. The interferometer is locked using a two-color scheme, where a far-detuned laser (blue) is used as a reference, and a feedback loop is implemented with a FPGA and a piezo-electric transducer (PZT). The low-power, resonant interference signal (red) is separated from the higher-power locking beam (blue) through a grating mirror. The filtered signal is then captured by a single-photon avalanche photodiode (SPAPD). PM_{res} and PM_{lock} are the two power meters used to stabilize the laser powers. (b) Evolution of the interference signal with detuning of the resonant laser (relative to the most pronounced QD transition) when the QD is tuned on (green). Same laser tuning range interference evolution when the QD is switched off (orange) through the application of an electric field across the QD (DC-Stark effect). A zoom-in of the blue area is presented in Fig. 2 (b).

the resulting interference signal, while the locking laser frequency stays fixed. We tune the resonance frequency of the QD with a voltage applied across the sample by virtue of the DC-Stark effect³¹, allowing us to compare the on- and off-resonance cases, respectively (See Figure 1(b)), and determine directly and accurately the phase shift induced by the QD (see the Method Section and the Supplemental materials for more details). Figure 2(a) and (b) presents two examples of signals at different laser detunings. Away from resonance (Figure 2(a)), no significant intensity and phase change are observed, meaning the change of the electric field itself does not affect the laser transmission, while near resonance (Figure 2(b)), the fringe contrast and phase changes when the QD is set to be resonant with the laser field. Through a single measurement, we can thus infer both the phase and intensity changes experienced by the light field due to the interaction with the QD. The results are presented in Fig.

FIG. 2. (a) and (b) Direct interferometric data with the emitter tuned *on* (green) and *off* (orange) resonance using the external electric field, for two different laser-emitter detunings (integration time of 100ms per point). The measurement points are plotted along with corresponding sinusoidal fits (solid line). The data in (b) correspond to the detuning area marked in blue in Fig. 1(b). (c) and (d) Extracted respective phase shift and transmission for the two dipoles, labeled 1 and 2. The solid lines correspond to the fit of the data to the theory.

2, where the phase (c) and intensity (d) spectra of the two dipole transitions of the QD neutral exciton, labeled (1) and (2), are displayed. We fit the phase and intensity data of both dipoles simultaneously (See Supplementary Information), and infer the maximal phase shifts to be $\phi_{max,1} = (-0.06 \pm 0.03)\pi$ and $\phi_{max,2} = (-0.19 \pm 0.03)\pi$ radians, respectively.

Next, we examine the saturation of the phase shift in order to investigate its nonlinear response to changes in the incoming laser power. We consider dipole transition (2). In Fig. 3(a), we show several spectra taken at different laser power levels and the corresponding fitting of the full saturation behavior (see Supplementary Information), which is fully consistent with the data presented. For each power level, we determine the maximum experimentally observed phase shift and investigate the nonlinear behavior as the QD saturates, see Fig. 3(b). By using the experimental parameters extracted previously, we estimate that the saturation happens at a mean photon flux of $n_c \sim 0.39$ photons interacting with the QD during its lifetime (See³⁰ and Supplementary Information), well below the single-photon level. **This should enable to observe a differential phase shift between single and two-photon components (also often called "nonlinear" phase**

FIG. 3. (a) Measurements of the phase response of the QD versus detuning and for different excitation powers. The solid lines are the fit to the theory of the overall data set. (b) Maximum measured experimental phase shift as a function of input power (measured at PM_{res} , see Fig. 1(a)). The colored points correspond to the data shown in (a). The full horizontal line indicates the calculated saturation power of the transition $P_{sat} = 0.15$ mW. The dashed line indicates the saturating power $P_{sat}^* = 0.14$ mW of the maximal phase shift (See Supplementary Information)

shift), such as measured in Ref.¹⁹, essential for the implementation of controlled quantum operations.

The experimentally extracted phase shifts are limited by the coupling efficiency and decoherence of the QD and future experiments on fully lifetime-limited QD transitions²⁴ should allow observing a phase shift approaching $\pi/2$. Going beyond this would even be possible in the setting of chiral quantum optics³² where directional coupling entails that the reflective "loss channel" can be strongly suppressed Fig. 4(a) schematically illustrates the isotropic and chiral cases, respectively. In the ideal chiral case, the maximum possible phase shift of π can be realized, the ultimate goal for quantum phase gates^{1,33,34}. In contrast, the transmitted intensity would be unchanged at resonance, see Fig. 4 (b), i.e. no photons are lost and the scattering is thereby deterministic in transmission. Such a single-photon response, however, would be undetectable in intensity measurements and therefore require the interferometric method demonstrated here. It is interesting to further exploit the unusual behavior of the phase response in the chiral geometry. When the input light intensity is increased, a very abrupt phase response is predicted (see Fig. 4(c)), unlike in the symmetric configuration. Indeed, towards saturation the transmission coefficient at resonance (which is real) changes from a negative value to a positive value, resulting in a sudden shift in the phase from π to 0. This may find applications as an all-optical phase-switch^{20,23,35}. Similarly a sharp transition can be found while varying the dephasing rate (see Fig. 4(d)), which means it may be applicable as an ultra-sensitive probe of environmental decoherence processes of the QD. Finally, we rediscover that the case of ideal directionality

FIG. 4. The top illustration shows the scattering configuration for an isotropic (Left) and a chiral (Right) coupled system. In the latter, the reflection and transmission decay rates (γ_r and γ_t respectively) differ. Correspondingly (a) and (b) shows the phase shift and transmission intensity for the isotropic and chiral coupling. (c) Maximal phase shift ϕ_{max} as a function of the driving Rabi frequency Ω for different directional coupling efficiencies $\beta_{dir} = [1$ (blue), 0.9 (red) 0.7 (yellow) 0.5 (purple)]. $\beta_{dir} = 0.5$ corresponds to the case of an isotropic waveguide with $\beta = 1$. (d) ϕ_{max} as a function of the pure dephasing rate γ_d for a series of coupling efficiency $\beta_{dir} = \{1$ (blue), 0.9 (red) 0.7 (yellow) 0.5 (purple)}

is equivalent to an ideal emitter in an isotropic waveguide when the efficiency decreases by half due to saturation ($\Omega \geq \frac{\gamma}{2\sqrt{2}}$), dephasing ($\gamma_{dp} \geq \gamma/2$), or coupling inefficiency ($\beta_{dir} \leq 1/2$).

In summary, we have developed an interferometric method for measuring the nonlinear phase shift of light caused by a single quantum emitter and measured an unprecedented phase response in a waveguide. These results may open up for a wide range of applications on how to realize deterministic quantum phase gates in photonic circuits^{33,36} as a basis for quantum non-demolition measurements^{17,18} or deterministic generation of optical Schrödinger cat states³⁷, when combined with accurate spin control^{34,38,39}. This work holds promises for on-chip photonic quantum processing, in particular combined with the recent achievement on the integration⁴⁰ and coherent coupling⁴¹ of multiple quantum dots in waveguides. Additionally, the quantum emitter phase shift may be applied as the quantum nonlinear operation required in quantum optical neural networks³, where even moderate nonlinear phase shifts have been shown to suffice for improving the implementation of Bell-state detectors^{11,12}. Finally, chiral light-matter interaction promises to improve the phase response even further,

although the combination of a high β -factor and high directionality has not yet explicitly been demonstrated in a waveguide. In such a configuration, interferometric measurements are required to detect the single-photon scattering processes, and the complex phase response acquired by optical pulses constitutes an interesting future direction of research that also may shed new light on applications of the emitter nonlinearity.

I. ACKNOWLEDGMENTS

We thank Vasiliki Angelopoulou for her help at the early stages of the experiment. We acknowledge funding from the Danish National Research Foundation (Center of Excellence “Hy-Q,” Grant No. DNR139) and from the European Union’s Horizon 2020 research and innovation programmes under Grant Agreements No. 824140 (TOCHA, H2020-FETPROACT-01-2018). This project has also received funding from BMBF 16KIQS009.

II. METHODS

Our quantum dot (QD) was embedded in a photonic crystal waveguide with a radius of 70 nm and a lattice constant of 250 nm. The sample configuration closely resembled the layout utilized in Reference²⁴. The bandgap was positioned approximately 0.5 THz away from the emission wavelength resulting in only weak Purcell enhancement while still maintaining a high β -factor²⁴. Light was coupled to and from the chip through shallow-etched grating couplers, where efficiencies of $> 25\%$, are typically reached while grating back reflections are strongly suppressed⁴².

The QD was tuned in and out of resonance through DC stark shift tuning using a voltage field. The “off” state (corresponding to an applied voltage of 0.8V) was checked to be away from any optically active transition of the dot. The “on” voltage was set to 1.24V. The linewidths of the two quantum dot dipole transitions were fitted to be 1.95 ± 0.05 GHz and 1.45 ± 0.05 GHz wide.

The interferometer is locked by having a piezoelectric transducer (PZT) mounted mirror to compensate for any change in phase not originating from the quantum emitter (see Supplemental Information for more details on the experimental setup). The feedback is performed by using an FPGA (Field Programmable Array, Red Pitaya) programmed to act like a lock-in amplifier followed by a proportional-integral-derivative controller⁴³.

The locking laser is blue-detuned by 7.5 nm from the QD transition at 941 nm to avoid any interaction with the emitter, and at a much higher power than the few-photon resonant laser. This wavelength was chosen to stay away from the QD transition while keeping a good transmission in the sample (away from the bandgap). We saw no difference compared to the use of a red-detuned laser. The response frequency of the mirror and piezoelectric

system is limited to 4kHz. The lock-in modulation signal

was chosen to be driven at 3.1kHz.

- ¹ D. E. Chang, V. Vuletić, and M. D. Lukin, *Nat. Photonics* **8**, 685 (2014).
- ² R. Uppu, L. Midolo, X. Zhou, J. Carolan, and P. Lodahl, *Nat. Nanotechnol.* **16**, 1308 (2021).
- ³ G. R. Steinbrecher, J. P. Olson, D. Englund, and J. Carolan, *npj Quantum Inf.* **5**, 1 (2019).
- ⁴ P. Lodahl, S. Mahmoodian, and S. Stobbe, *Rev. Mod. Phys.* **87**, 347 (2015).
- ⁵ P. Türschmann, H. Le Jeannic, S. F. Simonsen, H. R. Haakh, S. Götzinger, V. Sandoghdar, P. Lodahl, and N. Rotenberg, *Nanophotonics* **8**, 1641 (2019).
- ⁶ M. Arcari, I. Söllner, A. Javadi, S. Lindskov Hansen, S. Mahmoodian, J. Liu, H. Thyrestrup, E. H. Lee, J. D. Song, S. Stobbe, and P. Lodahl, *Phys. Rev. Lett.* **113**, 093603 (2014).
- ⁷ J.-T. Shen and S. Fan, *Phys. Rev. Lett.* **98**, 153003 (2007).
- ⁸ H. Le Jeannic, A. Tiranov, J. Carolan, T. Ramos, Y. Wang, M. H. Appel, S. Scholz, A. D. Wieck, A. Ludwig, N. Rotenberg, L. Midolo, J. J. García-Ripoll, A. S. Sørensen, and P. Lodahl, *Nat. Phys.* **18**, 1191 (2022).
- ⁹ D. Witthaut, M. D. Lukin, and A. S. Sørensen, *Europhys. Lett.* **97**, 50007 (2012).
- ¹⁰ F. Yang, M. M. Lund, T. Pohl, P. Lodahl, and K. Mølmer, *Phys. Rev. Lett.* **128**, 213603 (2022).
- ¹¹ A. Pick, E. S. Matekole, Z. Aqua, G. Guendelman, O. Firstenberg, J. P. Dowling, and B. Dayan, *Phys. Rev. Appl.* **15**, 054054 (2021).
- ¹² J. Ewaniuk, J. Carolan, B. J. Shastri, and N. Rotenberg, *Adv. Quantum Technol.* **6**, 2200125 (2023).
- ¹³ A. S. Zibrov, M. D. Lukin, L. Hollberg, D. E. Nikonov, M. O. Scully, H. G. Robinson, and V. L. Velichansky, *Phys. Rev. Lett.* **76**, 3935 (1996).
- ¹⁴ S. A. Aljunid, M. K. Tey, B. Chng, T. Liew, G. Maslennikov, V. Scarani, and C. Kurtsiefer, *Phys. Rev. Lett.* **103**, 153601 (2009).
- ¹⁵ M. Fischer, B. Srivathsan, L. Alber, M. Weber, M. Sondermann, and G. Leuchs, *Appl. Phys. B* **123**, 1 (2017).
- ¹⁶ Q. A. Turchette, C. J. Hood, W. Lange, H. Mabuchi, and H. J. Kimble, *Phys. Rev. Lett.* **75**, 4710 (1995).
- ¹⁷ A. Reiserer, S. Ritter, and G. Rempe, *Science* **342**, 1349 (2013).
- ¹⁸ J. Volz, R. Gehr, G. Dubois, J. Estève, and J. Reichel, *Nature* **475**, 210 (2011).
- ¹⁹ J. Volz, M. Scheucher, C. Junge, and A. Rauschenbeutel, *Nat. Photonics* **8**, 965 (2014).
- ²⁰ T. G. Tiecke, J. D. Thompson, N. P. de Leon, L. R. Liu, V. Vuletić, and M. D. Lukin, *Nature* **508**, 241 (2014).
- ²¹ I. Fushman, D. Englund, A. Faraon, N. Stoltz, P. Petroff, and J. Vučković, *Science* **320**, 769 (2008).
- ²² D. Wang, H. Kelkar, D. Martín-Cano, D. Rattenbacher, A. Shkarin, T. Utikal, S. Götzinger, and V. Sandoghdar, *Nat. Phys.* **15**, 483 (2019).
- ²³ M. Pototschnig, Y. Chassagneux, J. Hwang, G. Zumofen, A. Renn, and V. Sandoghdar, *Phys. Rev. Lett.* **107**, 063001 (2011).
- ²⁴ F. T. Pedersen, Y. Wang, C. T. Olesen, S. Scholz, A. D. Wieck, A. Ludwig, M. C. Löbl, R. J. Warburton, L. Midolo, R. Uppu, and P. Lodahl, *ACS Photonics* **7**, 2343 (2020).
- ²⁵ A. Javadi, I. Söllner, M. Arcari, S. L. Hansen, L. Midolo, S. Mahmoodian, G. Kiršanskė, T. Pregolato, E. H. Lee, J. D. Song, S. Stobbe, and P. Lodahl, *Nat. Commun.* **6**, 1 (2015).
- ²⁶ S. Faez, P. Türschmann, H. R. Haakh, S. Götzinger, and V. Sandoghdar, *Phys. Rev. Lett.* **113**, 213601 (2014).
- ²⁷ N. O. Antoniadis, N. Tomm, T. Jakubczyk, R. Schott, S. R. Valentin, A. D. Wieck, A. Ludwig, R. J. Warburton, and A. Javadi, *npj Quantum Inf.* **8**, 1 (2022).
- ²⁸ A. Pscherer, M. Meierhofer, D. Wang, H. Kelkar, D. Martín-Cano, T. Utikal, S. Götzinger, and V. Sandoghdar, *Phys. Rev. Lett.* **127**, 133603 (2021).
- ²⁹ A. Jechow, B. G. Norton, S. Händel, V. Blüms, E. W. Streed, and D. Kielpinski, *Phys. Rev. Lett.* **110**, 113605 (2013).
- ³⁰ H. Le Jeannic, T. Ramos, S. F. Simonsen, T. Pregolato, Z. Liu, R. Schott, A. D. Wieck, A. Ludwig, N. Rotenberg, J. J. García-Ripoll, and P. Lodahl, *Phys. Rev. Lett.* **126**, 023603 (2021).
- ³¹ G. Kiršanskė, H. Thyrestrup, R. S. Daveau, C. L. Dreeßen, T. Pregolato, L. Midolo, P. Tighineanu, A. Javadi, S. Stobbe, R. Schott, A. Ludwig, A. D. Wieck, S. I. Park, J. D. Song, A. V. Kuhlmann, I. Söllner, M. C. Löbl, R. J. Warburton, and P. Lodahl, *Phys. Rev. B* **96**, 165306 (2017).
- ³² P. Lodahl, S. Mahmoodian, S. Stobbe, A. Rauschenbeutel, P. Schneeweiss, J. Volz, H. Pichler, and P. Zoller, *Nature* **541**, 473 (2017).
- ³³ T. C. Ralph, I. Söllner, S. Mahmoodian, A. G. White, and P. Lodahl, *Phys. Rev. Lett.* **114**, 173603 (2015).
- ³⁴ J. Borregaard, A. S. Sørensen, and P. Lodahl, *Adv. Quantum Technol.* **2**, 1800091 (2019).
- ³⁵ W. Chen, K. M. Beck, R. Bücker, M. Gullans, M. D. Lukin, H. Tanji-Suzuki, and V. Vuletić, *Science* **341**, 768 (2013), <https://www.science.org/doi/pdf/10.1126/science.1238169>.
- ³⁶ Z. Chen, Y. Zhou, J.-T. Shen, P.-C. Ku, and D. Steel, *Phys. Rev. A* **103**, 052610 (2021).
- ³⁷ B. Wang and L.-M. Duan, *Phys. Rev. A* **72**, 022320 (2005).
- ³⁸ L.-M. Duan and H. J. Kimble, *Phys. Rev. Lett.* **92**, 127902 (2004).
- ³⁹ D. Tiarks, S. Schmidt, G. Rempe, and S. Dürr, *Sci. Adv.* **2**, e1600036 (2016).
- ⁴⁰ C. Papon, Y. Wang, R. Uppu, S. Scholz, A. D. Wieck, A. Ludwig, P. Lodahl, and L. Midolo, *Phys. Rev. Appl.* **19**, L061003 (2023).
- ⁴¹ A. Tiranov, V. Angelopoulou, C. J. van Diepen, B. Schriniski, O. A. D. Sandberg, Y. Wang, L. Midolo, S. Scholz, A. D. Wieck, A. Ludwig, A. S. Sørensen, and P. Lodahl, *Science* **379**, 389 (2023).
- ⁴² X. Zhou, I. Kulkova, T. Lund-Hansen, S. L. Hansen, P. Lodahl, and L. Midolo, *Appl. Phys. Lett.* **113** (2018), 10.1063/1.5055622.
- ⁴³ M. A. Luda, M. Drechsler, C. T. Schmiegelow, and J. Codrnia, *Review of Scientific Instruments* **90**, 023106 (2019), <https://doi.org/10.1063/1.5080345>.

Supplemental Material

Mathias J.R. Staunstrup,¹ Alexey Tiranov,¹ Ying Wang,¹ Sven Scholz,² Andreas D. Wieck,² Arne Ludwig,² Leonardo Midolo,¹ Nir Rotenberg,¹ Peter Lodahl,¹ and Hanna Le Jeannie¹

¹*Center for Hybrid Quantum Networks (Hy-Q), Niels Bohr Institute, University of Copenhagen, DK-2100 Copenhagen Ø, Denmark*

²*Lehrstuhl für Angewandte Festkörperphysik, Ruhr-Universität Bochum, Universitätsstraße 150, 44801 Bochum, Germany*

I. TRANSMISSION OF THE EMITTER-WAVEGUIDE SYSTEM

The quantum dot is modeled as a two-level system (TLS) with ground- and excited states $|g\rangle$ and $|e\rangle$. The Hamiltonian describing the light-emitter interaction can be written as¹:

$$\hat{H} = -\hbar\Delta\hat{\sigma}_{eg}\hat{\sigma}_{ge} + \hbar\omega_p\hat{\mathbf{f}}^\dagger(\mathbf{r})\hat{\mathbf{f}}(\mathbf{r}) - \hat{\mathbf{d}} \cdot \hat{\mathbf{E}}(\mathbf{r}) \quad (1)$$

The first term describes the emitter dynamics with $\Delta = \omega - \omega_{TLS}$ as the detuning between the driving field of frequency ω and the two-level system resonance ω_{TLS} . $\hat{\sigma}_{ij} = |i\rangle\langle j|$, where $i, j \in \{|g\rangle, |e\rangle\}$ are the transition operators of the TLS. The second term in the Hamiltonian accounts for the photon field at position \mathbf{r} with the bosonic annihilation operators $\hat{\mathbf{f}}(\mathbf{r})$. Finally, the last term accounts for the light-matter interaction between the emitter dipole $\hat{\mathbf{d}}$ and the electric field $\hat{\mathbf{E}}(\mathbf{r}) = \hat{\mathbf{E}}^+(\mathbf{r}) + \hat{\mathbf{E}}^-(\mathbf{r})$. The response of the TLS can be expressed by the partially traced density matrix giving the elements ρ_{ij} . In the rotating wave approximation and solving for the steady state solution ($\dot{\hat{\rho}} = 0$) we obtain the elements:

$$\begin{aligned} \rho_{ee} &= \frac{2\gamma_2\Omega^2}{\gamma(\gamma_2^2 + \Delta^2 + 4(\gamma_2/\gamma)\Omega^2)} \\ \rho_{ge} &= -\frac{\Omega(i\gamma_2 + \Delta)}{\gamma_2^2 + \Delta^2 + 4(\gamma_2/\gamma)\Omega^2} \end{aligned} \quad (2)$$

Where γ is the total emission rate that together with the pure dephasing rate γ_{dp} constitutes $\gamma_2 = \gamma/2 + \gamma_{dp}$. While the population is also dependent on the driving field amplitude through Rabi frequency $\Omega = \mathbf{d} \cdot \mathbf{E}/\hbar$.

In a single-mode conventional waveguide, the resulting transmitted "output" electric field can be expressed in terms of the input driving field^{1,2}:

$$\hat{\mathbf{E}}_{\text{out}}^+(\mathbf{r}) = \hat{\mathbf{E}}_{\text{in}}^+(\mathbf{r}) + i\frac{\beta\gamma}{2\Omega}\hat{\mathbf{E}}_{\text{in}}^+(\mathbf{r})\hat{\sigma}_{ge} \quad (3)$$

Where waveguide-emitter coupling efficiency is governed by the ratio $\beta = \frac{\gamma_{WG}}{\gamma}$. Here γ_{WG} is the rate of decay into the waveguide mode. The coupling factor is divided by 2 as equal coupling to both directions of propagation is assumed i.e. the coupling is isotropic. From this we define the corresponding transmission coefficient t that transforms the input electric field $\hat{\mathbf{E}}_{\text{in}}^+(\mathbf{r})$ to $\hat{\mathbf{E}}_{\text{out}}^+(\mathbf{r})$

through the photonic waveguide. Using equation 3, results in:

$$t = \frac{\langle \hat{\mathbf{E}}_{\text{out}}^+(\mathbf{r}) \rangle_{ss}}{\langle \hat{\mathbf{E}}_{\text{in}}^+(\mathbf{r}) \rangle_{ss}} = 1 + i\frac{\beta\gamma}{2\Omega}\rho_{eg} \quad (4)$$

Inserting the density matrix element $\rho_{eg} = \rho_{ge}^*$ of equation (2), we obtain:

$$t = 1 - \frac{\beta\gamma}{2} \frac{(\gamma_2 + i\Delta)}{\gamma_2^2 + \Delta^2 + 4(\gamma_2/\gamma)\Omega^2} \quad (5)$$

Finally, the normalized intensity of the transmitted light can be calculated as:

$$\begin{aligned} I_t &= \frac{\langle \hat{\mathbf{E}}_{\text{out}}^-(\mathbf{r})\hat{\mathbf{E}}_{\text{out}}^+(\mathbf{r}) \rangle_{ss}}{\langle \hat{\mathbf{E}}_{\text{in}}^-(\mathbf{r})\hat{\mathbf{E}}_{\text{in}}^+(\mathbf{r}) \rangle_{ss}} \\ &= 1 - \frac{\beta\gamma\gamma_2(2 - \beta)}{2(\gamma_2^2 + \Delta^2 + 4(\gamma_2/\gamma)\Omega^2)} \end{aligned} \quad (6)$$

We emphasize that $I_t \neq |t|^2$.

A. Maximal phase shift

The maxima of the phase shift with respect to the detuning can be found, at low power ($\Omega \ll 1$) and in the absence of dephasing, by solving:

$$\frac{\partial \arg(t)}{\partial \Delta}(\Delta_{\pm}) = \frac{2\beta\gamma((\beta - 1)\gamma^2 + 4\Delta_{\pm}^2)}{(\gamma^2 + 4\Delta_{\pm}^2)((\beta - 1)^2\gamma^2 + 4\Delta_{\pm}^2)} = 0 \quad (7)$$

which corresponds to

$$\Delta_{\pm} = \pm\gamma\frac{\sqrt{1 - \beta}}{2} \quad (8)$$

Plugging this back in the expression of the argument, one can find :

$$\begin{aligned} |\phi|_{max} &= |\arg(t(\Delta_{\pm}))| \\ &= \arg\left(\frac{(2 - i\sqrt{1 - \beta})\beta - 2}{\beta - 2}\right) \\ &= \tan^{-1}\left(\frac{\beta}{2\sqrt{1 - \beta}}\right) \end{aligned} \quad (9)$$

B. Transmission for a chirally coupled emitter

In a waveguide with chiral light-matter coupling the interaction is directionally dependent. Similar to before, the total electric field in transmission is

$$\hat{\mathbf{E}}_{\text{out}}^+(\mathbf{r}) = \hat{\mathbf{E}}_{\text{in}}(\mathbf{r}) + i \frac{\beta_{\text{dir}} \gamma}{\Omega} \hat{\mathbf{E}}_{\text{in}}^+(\mathbf{r}) \hat{\sigma}_{ge} \quad (10)$$

where we define the directional coupling efficiency as $\beta_{\text{dir}} = \gamma_t/\gamma$, by differing the emission rate in transmitted(t) or reflected modes(r). Following the same method as for conventional waveguide, we have:

$$t_{\text{dir}} = 1 - \frac{\beta_{\text{dir}} \gamma (\gamma_2 + i\Delta)}{\gamma_2^2 + \Delta^2 + 4(\gamma_2/\gamma)\Omega^2} \quad (11)$$

$$I_{t_{\text{dir}}} = 1 + \frac{2\beta_{\text{dir}} \gamma \gamma_2 (\beta_{\text{dir}} - 1)}{\gamma_2^2 + \Delta^2 + 4(\gamma_2/\gamma)\Omega^2}$$

Note that in the case of an isotropic, conventional waveguide ($\gamma_t = \gamma_r = \gamma_{WG}/2$), we recover the equation for an emitter coupled isotropically to waveguide modes.

II. MACH-ZEHNDER INTERFEROMETRY

The intensity of the output modes in a Mach-Zehnder interferometer is affected by the difference in phase, $\delta\phi$, between the two paths in the interferometer:

$$I = \sin^2(\delta\phi/2) \quad (12)$$

When light at frequency f travels through each arm of the Mach-Zehnder interferometer (1,2), it experiences a phase shift of $\phi_{1,2} = 2\pi f L_{1,2}/cn_{1,2}$, where the speed of light is c and the index of refraction n may be different in the two arms with respective distances $L_{1,2}$. Additionally, there may be an environmental fluctuation phase difference $\delta\phi_{\text{env}}$. Only one path (path 1) is affected by a phase change $\phi_{QD} = \arg(t)$ induced by the quantum dot waveguide system. Therefore, the final interferometric phase difference can be expressed as

$$\delta\phi = \phi_1 - \phi_2 = \frac{2\pi f \delta L}{c} + \delta\phi_{\text{env}} + \phi_{QD} \quad (13)$$

δL is the interferometric path length difference. The interferometric signal obtained when sweeping the laser detuning is displayed in Fig. 1(b). The Fourier transform(FFT) of these interferometric fringes is displayed in Fig. SM SM1. Using Equations 12 and 13, we identify the main frequency component of the Fourier transform as $f = \delta L/c$ and we estimate the full path length difference of our interferometer to be $\delta L \approx 2.78m$.

The interferometer was stabilized by mounting a mirror in one of the Mach-Zehnder arms on a piezoelectric transducer(PZT). This allowed upon the application of a voltage to modulate the optical path length difference and thus correct for any fluctuation or drift. This Mirror/PZT system was found to have its first frequency

Fig. SM 1: Normalized Fourier transformation of the interferometric signal as function of δL in meters with the QD turned off.

Fig. SM 2: Normalized and background subtracted fringe signal seen on the oscilloscope of the modulated interferometer(blue) and output of the stabilized interferometer (black)

harmonic at around 4KHz. By modulating the PZT voltage at 3.1 KHz with a small amplitude, a lock-in amplifier was used to gain a signal that looks roughly like the first derivative of the interferometric signal. Using this as the error signal for feedback PID loop to lock at the zero value of the lock-in output resulted in a top-of-fringe locking³. FIG.SM2 show in blue the normalized fringes signal recorded by modulating the PZT with a ramp signal. In the recorded range, the signal shows that the mirror was displaced roughly 4 wavelengths. Compared to this in black is the top-of-fringe locked signal taken in a subsequent measurement.

Fig. SM 3: Visualization of the effect of transition 2 (colored dots), and its corresponding fit (dashed black line) in a phasor diagram as a function of the normalized detuning Δ/γ . For comparison, solid colored lines from inner to outer curves represent calculations for increasing directional coupling efficiencies $\beta_{dir} = \{0.5, 0.8, 1\}$.

III. MODELING THE EXPERIMENTAL DATA

A. Phase and Intensity

We simultaneously fit the phase and intensity data of the two dipoles' response displayed in Fig. 2(c) and (d). We assume here for simplicity identical dephasing rates for both dipoles. Furthermore, we assume only pure dephasing, while in reality also slow noise processes (spectral diffusion) are influencing, however an unambiguous separation of these two processes is outside the scope of the present work; for more information, see⁴. As a consequence, the extracted pure dephasing rates will be overestimated. We adjusted the displayed data by taking into account the constant offset ϕ_0 caused by weak Fano resonances, which are a result of partial reflection from the outcoupling gratings of the waveguide. (More information can be found in the references^{4,5}) We find the parameters to be:

	Dipole 1	Dipole 2
β	0.94 ± 0.03	1
γ (ns ⁻¹)	9.4 ± 0.2	12.3 ± 0.2
γ_{dp} (ns ⁻¹)	3.9 ± 0.1	
ϕ_0 (rad)	-0.25 ± 0.02	

We present the data of dipole transition 2, and the corresponding model fit in a phasor diagram in Fig. SM 3

B. Saturation Characterization

In the following, we focus only on transition (2). We fit all the transmission spectra series with power presented in Fig. 3 simultaneously using a nonlinear least-squares regression based on our model:

$$\phi(\Delta, \Omega) = \arg \left[1 - \frac{\beta\gamma(\gamma_2 + i\Delta)}{2(\Delta^2 + \gamma_2^2 + 4\Omega^2\gamma_2/\gamma)} \right] \quad (14)$$

Since we do not know the exact Rabi frequency we define the mapping efficiency constant η to the power as $\Omega = \sqrt{\eta P}$. We obtain the single set of parameters:

β	0.99 [0.57, 1]
γ (ns ⁻¹)	12.6 [7.7, 17.4]
γ_{dp} (ns ⁻¹)	3.4[0, 7.4]
ϕ_0 (rad)	-0.26 [-0.31, -0.2]
η (rad.s ⁻¹ .mW ⁻¹)	5 [2.3, 7.7]

Those values are in good agreement with the data of the two dipoles in Figure 2. At resonance, the saturation parameter is defined as $S = \frac{4\Omega^2}{\gamma\gamma_2}$ (where $\rho_{ee} = \frac{S}{2(S+1)} = 1/4$ at $S = 1$). By using the fitted parameters, this gives: $P_{sat} = \frac{\gamma(\gamma/2 + \gamma_{dp})}{4\eta} \approx 0.15$ mW (indicated by a full line in Fig. 3(b)). Similarly, we can estimate the critical photon flux from the fitted parameters as: $n_c = \frac{1+2\beta\gamma_{dp}/\gamma}{4\beta^2} \sim 0.39[0.25, 0.73]$ ^{4,5}. This is comparable to previously measured values for solid states: quantum dots: $n_c \sim [0.81^5, 0.33^4, 1.6^6]$ in weak coupling and $n_c \sim 0.4 - 0.6$ in strong coupling⁷; and Organic Molecules: $n_c \sim 25^8$ in weak coupling and $n_c \sim 0.44^9$ in strong coupling. An atom strongly coupled to a resonator has demonstrated even lower values, with $n_c \sim 0.02^{10}$. At each power we also perform an independent fit with free parameters, to extract accurately the maximal, experimentally measured phase shift from the data. Those are the data points $|\phi_{max}|$ displayed in Fig.3(b). By directly fitting by a general decay model $|\phi_{max}| = Ae^{P/P_{sat}^*} + B$, we find $P_{sat}^* = 0.14$ mW [0.11, 0.16]. This P_{sat}^* is the saturation power of the maximal phase shift, and is indicated by a dotted line in Fig. 3(b).

IV. QD AND WAVEGUIDE SYSTEM

Initial experimental measurements consisted of investigating the transmission response of the photonic crystal waveguide and identifying an optical transition from a quantum dot embedded in the waveguide. A widefield image of a waveguide system leading through a photonic crystal waveguide crystal is shown in FIG.SM4(a). Here the light can be coupled in/out of the waveguide via the couplers at the ends. The larger area along the waveguide is the photonic crystal consisting of a regular lattice with a periodicity of 250nm with hole sizes of 70nm. By collecting

the transmitted signal we can obtain the transmission response by scanning the laser frequency. The resulting transmission response is shown in FIG.SM4(b). The red and blue lines show respectively the laser frequency of the on-resonance laser and locking laser. When the band of the transmission was known, finer ranges of frequencies were scanned until the transmission dip of a quantum dot was found. By Scanning the frequency for different voltages we are able to build the transmission map as shown in FIG.SM5. The voltage for the experiment was set at 1.24V for the "on" mode i.e. we have turned on the optical transition. In contrast, the "off" mode had the voltage set to 0.8V sufficiently far from an optical transition.

Fig. SM 4: (a) Widefield image of the sample with the photonic waveguide, with the laser spot next to it. (b) Unnormalized transmission spectra of the laser as a function of the wavelength in the waveguide. The bandgap can be easily localized and is around 318.3 THz. The red and blue line marks the frequency of the on-resonance and locking laser, respectively.

Fig. SM 5: Normalized transmission of the laser through the waveguide as function of the frequency and the voltage applied on the sample. The two dipole transitions can be identified.

- ¹ P. Türschmann, H. Le Jeannic, S. F. Simonsen, H. R. Haakh, S. Götzinger, V. Sandoghdar, P. Lodahl, and N. Rotenberg, *Nanophotonics* **8**, 1641 (2019).
- ² A. Asenjo-Garcia, J. D. Hood, D. E. Chang, and H. J. Kimble, *Phys. Rev. A* **95**, 033818 (2017).
- ³ M. A. Luda, M. Drechsler, C. T. Schmiegelow, and J. Codrnia, *Review of Scientific Instruments* **90**, 023106 (2019), <https://doi.org/10.1063/1.5080345>.
- ⁴ H. Le Jeannic, T. Ramos, S. F. Simonsen, T. Pregonolato, Z. Liu, R. Schott, A. D. Wieck, A. Ludwig, N. Rotenberg, J. J. García-Ripoll, and P. Lodahl, *Phys. Rev. Lett.* **126**, 023603 (2021).
- ⁵ A. Javadi, I. Söllner, M. Arcari, S. L. Hansen, L. Midolo, S. Mahmoodian, G. Kiršanskė, T. Pregonolato, E. H. Lee, J. D. Song, S. Stobbe, and P. Lodahl, *Nat. Commun.* **6**, 1 (2015).
- ⁶ H. Thyrrstrup, G. Kirsanske, H. Le Jeannic, T. Pregonolato, L. Zhai, L. Raahauge, L. Midolo, N. Rotenberg, A. Javadi, R. Schott, A. D. Wieck, A. Ludwig, M. C. Löbl, I. Söllner, R. J. Warburton, and P. Lodahl, *Nano Letters* **18**, 1801 (2018).
- ⁷ I. Fushman, D. Englund, A. Faraon, N. Stoltz, P. Petroff, and J. Vučković, *Science* **320**, 769 (2008).
- ⁸ M. Pototschnig, Y. Chassagneux, J. Hwang, G. Zumofen, A. Renn, and V. Sandoghdar, *Phys. Rev. Lett.* **107**, 063001 (2011).
- ⁹ D. Wang, H. Kelkar, D. Martin-Cano, D. Rattenbacher, A. Shkarin, T. Utikal, S. Götzinger, and V. Sandoghdar,

- Nat. Phys.* **15**, 483 (2019).
¹⁰ J. Volz, M. Scheucher, C. Junge, and A. Rauschenbeutel,
Nat. Photonics **8**, 965 (2014).

[H]

REVIEWERS' COMMENTS

Reviewer #3 (Remarks to the Author):

Thank you for the detailed response. I am now satisfied with the latest revisions.

I am happy to recommend this manuscript for publication.